# Correlation between leukocyte phenotypes and prognosis of amyotrophic lateral sclerosis

Can Cui[1]*, Caroline Ingre[2,3], Li Yin[4], Xia Li[4], John Andersson[1], Christina Seitz[1], Nicolas Ruffin[3], Yudi Pawitan[4], Fredrik Piehl[2,3], Fang Fang[1]*

[1]Unit of Integrative Epidemiology, Institute of Environmental Medicine, Karolinska Institutet, Stockholm, Sweden; [2]SLL- ME Neurologi, Karolinska University Hospital, Stockholm, Sweden; [3]Department of Clinical Neuroscience, Karolinska Institutet, Stockholm, Sweden; [4]Department of Medical Epidemiology and Biostatistics, Karolinska Institutet, Stockholm, Sweden

*For correspondence:
can.cui@ki.se (CC);
fang.fang@ki.se (FF)

**Competing interest:** The authors declare that no competing interests exist.

**Abstract** The prognostic role of immune cells in amyotrophic lateral sclerosis (ALS) remains undetermined. Therefore, we conducted a longitudinal cohort study including 288 ALS patients with up to 5-year follow-up during 2015–2020 recruited at the only tertiary referral center for ALS in Stockholm, Sweden, and measured the levels of differential leukocytes and lymphocyte subpopulations. The primary outcome was risk of death after diagnosis of ALS and the secondary outcomes included functional status and disease progression rate. Cox model was used to evaluate the associations between leukocytes and risk of death. Generalized estimating equation model was used to assess the correlation between leukocytes and functional status and disease progression rate. We found that leukocytes, neutrophils, and monocytes increased gradually over time since diagnosis and were negatively correlated with functional status, but not associated with risk of death or disease progression rate. For lymphocyte subpopulations, NK cells (HR= 0.61, 95% CI = [0.42–0.88] per SD increase) and Th2-diffrentiated CD4+ central memory T cells (HR= 0.64, 95% CI = [0.48–0.85] per SD increase) were negatively associated with risk of death, while CD4+ effector memory cells re-expressing CD45RA (EMRA) T cells (HR= 1.39, 95% CI = [1.01–1.92] per SD increase) and CD8+ T cells (HR= 1.38, 95% CI = [1.03–1.86] per SD increase) were positively associated with risk of death. None of the lymphocyte subpopulations was correlated with functional status or disease progression rate. Our findings suggest a dual role of immune cells in ALS prognosis, where neutrophils and monocytes primarily reflect functional status whereas NK cells and different T lymphocyte populations act as prognostic markers for survival.

## Editor's evaluation

Cui et al. colleagues carried out a longitudinal analysis of blood cell counts in a cohort of ALS patients and found increased numbers of neutrophils and monocytes, which negatively correlated with ALSFRS-R score, but not with rate of disease progression. They also found increased levels in NK and central memory TH2 T cells, which correlated with a lower risk of death. In contrast, increased levels of CD4 CD45RA effector memory and CD8 T cells were correlated with a higher risk of death. These findings have important implications for the pathogenesis of ALS as well as the development of immune-based ALS therapies targeting specific populations immune cells.

## Introduction

Amyotrophic lateral sclerosis (ALS) is a rare but devastating neurodegenerative disease. Although a genetic cause has been demonstrated for some cases of ALS, the etiology remains unknown for most of the patients with ALS. There is currently no cure or effective treatment available for ALS. A range of potential disease mechanisms have however been proposed, with potential for informing on novel therapeutic targets (*Zarei et al., 2015*).

Neuroinflammatory features, including local glial activation and T-cell infiltration in the central nervous system (CNS), are well documented in ALS (*Komine and Yamanaka, 2015*). Animal studies have demonstrated that altering the function of microglia and infiltrating T cells to the CNS affect disease progression in experimental ALS (*Beers et al., 2006*; *Martínez-Muriana et al., 2016*; *Beers et al., 2008*). Human studies have also documented signs of systemic immune activation in ALS patients, compared with healthy controls, suggesting that peripheral immune activation may play a role in human ALS through the peripheral–central neuroimmune crosstalk (*Liu et al., 2020*; *Jin et al., 2020*). Most of the previous studies (*Jin et al., 2020*; *Rentzos et al., 2012*; *Henkel et al., 2012*), however, are not population based and have limited sample size, raising the concern of potentially insufficient internal (e.g., selection bias and chance finding) and external (e.g., lack of generalizability) validity. Further, few studies have recorded immune cells longitudinally after ALS diagnosis, as most previous studies relied on a single measurement (*Sheean et al., 2018*).

The main purpose of this study was to determine cellular immune changes occurring over time since diagnosis of ALS and their prognostic values in a large clinical sample. For this purpose, we enrolled a longitudinal cohort of ALS patients, representing a large proportion of the entire ALS population in Stockholm, Sweden, to (1) describe the temporal changes of peripheral leukocytes over time since ALS diagnosis, (2) assess the association of immune cell dynamics with the risk of death after ALS diagnosis, and (3) evaluate the correlations of different cell populations with the functional status and disease progression rate of ALS.

## Results

### Leukocytes and lymphocyte subtypes in ALS

*Supplementary file 2* shows the distribution of leukocyte populations ($N$ = 288 patients) and lymphocyte subpopulations ($N$ = 92 patients) across all measures after ALS diagnosis. The vast majority of the cell populations were within the normal range, except for CD8$^+$ central memory (CM) cells, CD4$^+$HLA-DR$^+$CD38$^+$ cells and CD8$^+$HLA-DR$^+$CD38$^+$ cells, which were above the normal range.

In the main cohort, the levels of leukocytes, neutrophils, and monocytes increased progressively over time, especially from 20 months after diagnosis onward (*Figure 1*). These trends were statistically significant, with or without adjustment for age and sex, and remained statistically significant after correction for multiple testing (*Table 1*) . In contrast, no clear temporal trend was noted for lymphocytes. These results remained largely similar when stratifying the patients by sex, site of onset, or presence of *C9orf72* expansions (*Figure 1—figure supplement 1*). The levels of leukocytes, neutrophils, and monocytes increased, whereas the levels of lymphocytes decreased, after Riluzole treatment, compared with before such treatment (*Figure 1—figure supplement 2*).

In the FlowC cohort, no clear temporal trend was noted for any lymphocyte subpopulation, although ALS patients demonstrated persistently higher proportions of CD8$^+$ CM, CD4$^+$HLA-DR$^+$CD38$^+$, and CD8$^+$HLA-DR$^+$CD38$^+$ cells than the reference ranges (*Figure 1—figure supplement 3*). After adjustment for age and sex, there was a decreasing % of naive CD4$^+$ T cells whereas increasing %s of CD4$^+$ EMRA, CD4$^+$HLA-DR$^+$CD38$^-$, and CD8$^+$HLA-DR$^+$CD38$^-$ cells since ALS diagnosis (*Supplementary file 3*). Male patients showed lower levels of CD4$^+$, naive CD4$^+$, and Th2 of CD4$^+$ CM cells, but higher levels of CD4$^+$ effector memory (EM, CD4$^+$ EMRA, CD8$^+$, CD4$^+$HLA-DR$^+$CD38$^-$, and CD4$^+$HLA-DR$^+$CD38$^+$ cells, compared with female patients, especially early stage after diagnosis (*Figure 1—figure supplement 4*)). Patients with limb onset had lower levels of CD8$^+$ CM and CD4$^+$HLA-DR$^+$CD38$^+$ cells compared with patients with other site of onset, whereas carriers of *C9orf72* expansions had higher levels of natural killer (NK) cells and T cells, but lower levels of naive CD4$^+$ T cells and CD8$^+$ EM cells, than other patients (*Figure 1—figure supplement 5*).

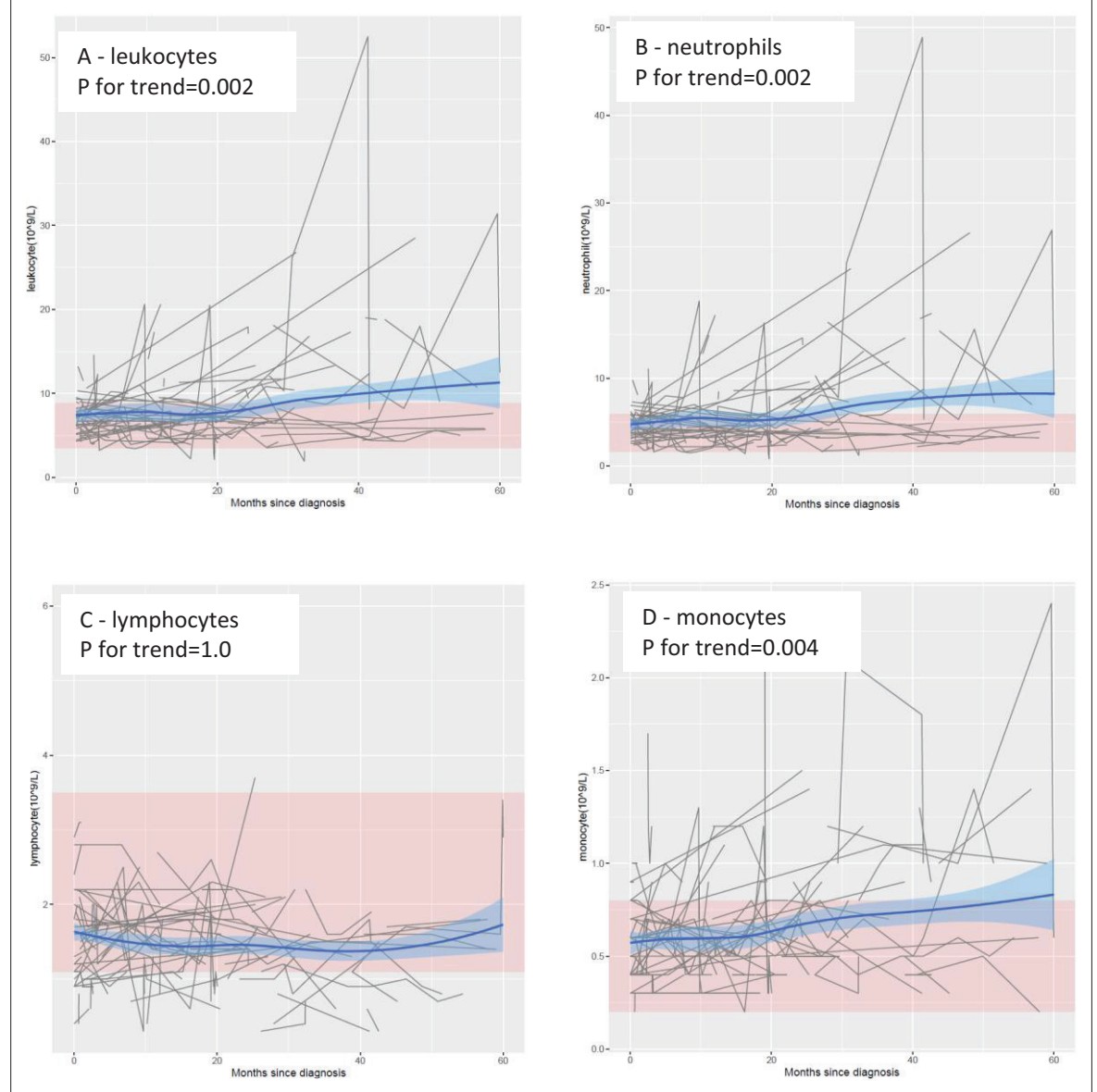

**Figure 1.** Mean levels of leukocyte populations after a diagnosis of amyotrophic lateral sclerosis (ALS). The black lines show measured levels of leukocyte populations for each patient. The blue lines and shadow areas show the mean levels of leukocyte populations with 95% confidence intervals. Pink areas indicate normal range. p for trend shows the p value of within-individual temporal change of each cell population after taking into account the relatedness of repeated measurements.

The online version of this article includes the following source data and figure supplement(s) for figure 1:

**Source data 1.** Levels of leukocyte populations from 3months before diagnosis of amyotrophic lateral sclerosis (ALS) onwards.

**Figure supplement 1.** Temporal trend of leukocyte populations by sex, site of onset, and presence of C9orf72 expansions.

**Figure supplement 2.** Temporal trend of leukocyte populations before and after Riluzole treatment.

**Figure supplement 3.** Lymphocyte populations that differed from normal range.

**Figure supplement 4.** Temporal trend of lymphocyte populations by sex.

**Figure supplement 5.** Temporal trend of lymphocyte populations that differed by site of onset and presence of C9orf72 expansions.

## Survival

During a median follow-up of 1.1 years, we observed 163 deaths or use of invasive ventilation among the 288 patients of the main cohort. No association was noted between the level of leukocytes, neutrophils, lymphocytes, or monocytes with risk of death (*Supplementary file 4*). This result did

**Table 1.** Temporal changes of leukocyte populations after diagnosis of amyotrophic lateral sclerosis (ALS), a cohort study of 288 patients with ALS in Stockholm, Sweden†.

| Cell type | Unadjusted | | | Adjusted* | | |
|---|---|---|---|---|---|---|
| | Coefficient | p value | FDR | Coefficient | p value | FDR |
| Leukocyte (10⁹/l) | 0.19 | **0.01** | **0.01** | 0.22 | **2.4E−03** | **4.7E−03** |
| Neutrophil (10⁹/l) | 0.18 | **3.6E−03** | **0.01** | 0.21 | **1.5E−03** | **4.7E−03** |
| Lymphocyte (10⁹/l) | 3.7E−03 | 0.73 | 0.73 | 4.1E−05 | 1.00 | 1.00 |
| Monocyte (10⁹/l) | 0.01 | **0.03** | **0.04** | 0.01 | **4.2E−03** | **0.01** |

Bold values denote statistical significance of p < 0.05.

*Adjusted for age at diagnosis and sex.

†Linear mixed model was applied to derive the coefficient estimates, per year and p value for trend.

‡

FDR: false discovery rate.

not change after excluding patients diagnosed before the start of the Motor Neuron Disease (MND) Quality Registry, focusing on first cell measure only, or excluding patients with *C9orf72* expansions (*Supplementary file 5*). Per standard deviation (SD) increase of neutrophil-to-lymphocyte ratio (NLR), there was a higher risk of death (hazard ratio [HR] = 1.31, 95% confidence interval [CI]: 1.13–1.52).

In the FlowC cohort, we found higher NK cell counts and %s of Th2-diffrentiated CD4⁺ CM cells to be associated with lower risk of death, whereas higher %s of CD4⁺ EMRA cells and CD8⁺ T cells were associated with higher risk of death (*Figure 2*). These results were largely similar after restricting the analysis to first measure of lymphocytes (data not shown) or after excluding patients with *C9orf72* expansions (*Figure 2—figure supplement 1*).

## Functional status and disease progression rate

In the main cohort, a higher level of leukocytes, neutrophils, or monocytes was associated with a lower Amyotrophic Lateral Sclerosis Functional Rating Scale – revised (ALSFRS-R) score measured at the time of sampling, whereas no such correlation was evident for lymphocyte counts (*Table 2*). There was, however, no association of leukocytes, neutrophils, lymphocytes, or monocytes with disease progression rate. Disease progression rate differed however between patients with different NLRs (e.g., p = 0.04, comparing group with above third tertile NLR to the group with below first tertile NLR). A greater longitudinal increase of leukocytes, neutrophils, and monocytes was associated with a greater longitudinal decline in ALSFRS-R score (*Table 3*).

In the FlowC cohort, none of the lymphocyte subtypes was associated with ALSFRS-R score or disease progression rate measured at the time of sampling (*Supplementary file 6*).

## Sensitivity analysis for ongoing infection

We excluded 15 patients in the main cohort who had been sampled with the presence of infection. The results on risk of death, ALSFRS-R score and disease progression rate remained similar although some results lost statistical significance (*Supplementary files 7 and 8*). There was no patient with sampling during ongoing infection in the FlowC cohort.

## Discussion

We here report a longitudinal cohort study of temporal dynamics of white blood cell populations in 288 ALS patients in Stockholm, Sweden. We found higher counts of blood leukocytes, neutrophils, and monocytes to be associated with a lower functional status, but not with the risk of death after ALS diagnosis or disease progression rate. In a subsample of 92 patients, we found that higher NK cell counts and proportions of Th2-diffrentiated CD4⁺ T cells were associated with a lower risk of death, whereas higher counts of CD8⁺ T cells and proportions of CD4⁺ EMRA T cells were associated with higher mortality risk.

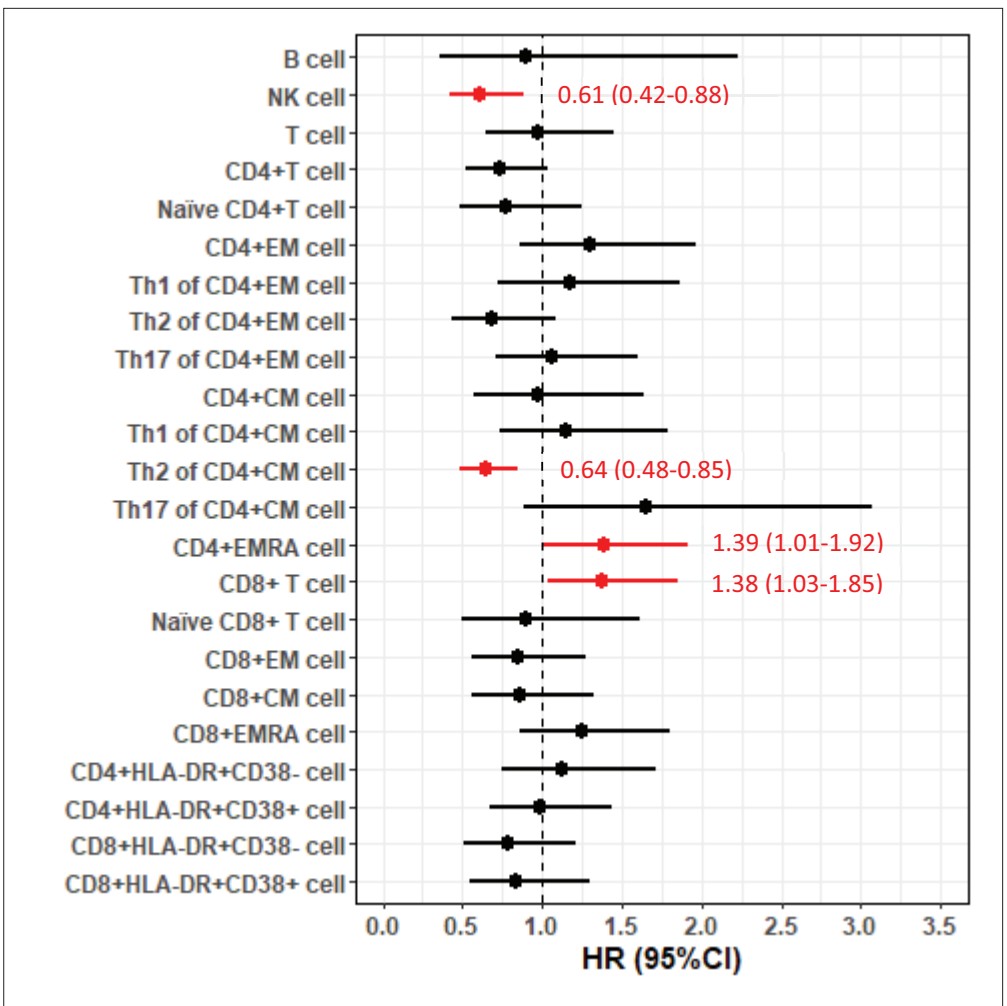

**Figure 2.** Forest plot of hazard ratios (HRs) and 95% confidence intervals (CIs) for the associations of lymphocyte populations with risk of death after a diagnosis of amyotrophic lateral sclerosis (ALS).

The online version of this article includes the following source data and figure supplement(s) for figure 2:

**Source data 1.** Hazard ratios (HRs) and 95% confidence intervals (CIs) for the associations of lymphocyte populations with risk of death after a diagnosis of amyotrophic lateral sclerosis (ALS).

**Figure supplement 1.** The associations of lymphocyte populations with risk of death after a diagnosis of amyotrophic lateral sclerosis (ALS), after excluding patients with C9orf72 expansions.

ALS patients demonstrated gradually increasing counts of leukocytes, neutrophils, and monocytes over time since diagnosis. Benjamin et al. also reported increasing counts of leukocytes and neutrophils in ALS patients (*Murdock et al., 2017*). Accumulating monocytes have also been shown in the cervical and lumbar spinal cord of ALS model over time (*Zondler et al., 2016*). The finding that peripheral leukocytes, neutrophils, and monocytes were correlated to ALSFRS-R, but not progression rate or survival, indicates that these leukocyte subtypes may serve better as markers for functional status than prognosis. A previous study also found CD16 expression on neutrophils and nonclassical monocytes to correlate with ALS disease severity (*McGill et al., 2020*). As neutrophils and monocytes have phagocytic function, increased levels of circulating neutrophils and monocytes might indicate enhanced muscle damage, explaining the correlation with deteriorating functional status. It has also been reported that circulating monocytes from ALS patients preferentially differentiate to a M1 proinflammatory phenotype and produce more interleukin 6 and tumor necrosis factor α, compared with monocytes from healthy controls (*Du et al., 2020*). Regardless, although it has been proposed that peripheral neutrophils and monocytes are recruited to the CNS through a disrupted brain–blood barrier (BBB) and affect the disease progression of experimental ALS through secreting

**Table 2.** Cross-sectional correlations between leukocyte populations and amyotrophic lateral sclerosis (ALS) Functional Rating Scale-revised (ALSFRS-R) score and disease progression rate, a cohort study of 288 ALS patients in Stockholm, Sweden[*].

| Cell type | ALSFRS-R | | | Progression rate | | |
|---|---|---|---|---|---|---|
| | Coefficient | p value | FDR | Coefficient | p value | FDR |
| Leukocyte (10⁹/l) | −2.80 | **4.0E−03** | **0.01** | 0.02 | 0.74 | 0.74 |
| Neutrophil (10⁹/l) | −3.10 | **1.0E−03** | **4.0E−03** | 0.05 | 0.33 | 0.67 |
| Lymphocyte (10⁹/l) | 1.48 | 0.15 | 0.15 | −0.08 | 0.32 | 0.67 |
| Monocyte (10⁹/l) | −2.75 | **2.0E−03** | **4.0E−03** | −0.03 | 0.52 | 0.69 |

Bold values denote statistical significance of p < 0.05.

[*]Generalized estimating equation model was applied to derive the coefficient estimates and p values, with adjustment for age at diagnosis and sex. ALSFRS-R score ranges from 0 to 48, with higher score showing better motor function status. Progression rate indicates the decline of motor function per month.

FDR: false discovery rate.

proinflammatory cytokines and influencing other cells (*Murdock et al., 2015*; *Butovsky et al., 2012*; *Zamudio et al., 2020*), the functional relevance in human ALS is still unclear (*Zondler et al., 2016*; *Ajami et al., 2007*).

NLR has been considered a prognostic marker in many diseases, such as cancer (*Akinci Ozyurek et al., 2017*; *Bowen et al., 2017*) and cardiovascular disease (*Kim et al., 2018*), which may indicate the involvement of complex coordination between different immune response pathways in the disease progression. Previous studies have also demonstrated a negative association between NLR and survival in ALS patients (*Choi et al., 2020*; *Wei et al., 2022*). Our study also found that a higher NLR was associated with a higher risk of death. However, the relation between NLR and disease progression rate of ALS is less consistently reported between studies (*Choi et al., 2020*; *Wei et al., 2022*), although we did find a higher NLR to be associated with a greater disease progression rate.

We found a higher level of blood-borne NK cells to be associated with a lower risk of death after ALS diagnosis. NK cells are considered a critical part of the innate immunity and function through lysing infected, oncogenic, apoptotic, and major histocompatibility complex (MHC) class I-deficient cells (*Vivier et al., 2011*). Although NK cells are known to penetrate BBB and interact with microglia, astroglia, and neurons (*Shi et al., 2011*), the role of NK cells in disease progression of ALS has rarely been addressed. NK cells may exert a cytotoxic role in the brain by its natural function, as suggested by a study showing that depletion of NK cells prolonged survival in ALS mouse models (*Garofalo et al., 2020*; *Murdock et al., 2021a*). As infiltration of NK cells to CNS may lead to a decrease in peripheral NK populations, the protective role of higher peripheral NK cells against risk of death, as observed in the present study, might be partly attributable to a lower level of its infiltration to the

**Table 3.** Associations between longitudinal changes in cell measures and longitudinal changes in Amyotrophic Lateral Sclerosis Functional Rating Scale – revised (ALSFRS-R) score, a cohort study of 288 patients with ALS in Stockholm, Sweden[†].

| Cell type | Unadjusted | | Adjusted* | |
|---|---|---|---|---|
| | Coefficient | p value | Coefficient | p value |
| Leukocyte (10⁹/l) | −5.72 | **0.010** | −5.41 | **0.012** |
| Neutrophil (10⁹/l) | −4.05 | **0.020** | −3.85 | **0.023** |
| Lymphocyte (10⁹/l) | −0.49 | 0.839 | −0.22 | 0.925 |
| Monocyte (10⁹/l) | −12.90 | **0.001** | −12.14 | **0.001** |

Bold values denote statistical significance of p < 0.05.

*Adjusted for age at diagnosis and sex.

[†]Generalized estimating equation model was applied to derive the coefficient estimates and p values, per unit change of log-transformed leukocyte counts.

CNS. On the other hand, NK cells may indeed also play a neuroprotective role. A previous study showed that lower levels of CD56[bright] NK cells in the CSF were associated with faster progression in ALS patients (*Rolfes et al., 2021*). Additionally, findings from experimental autoimmune encephalomyelitis, a model of multiple sclerosis, showed NK cells to suppress neuroinflammation, diminish tissue damage, and protect motoneurons (*Huang et al., 2006*; *Hammarberg et al., 2000*). Patients with multiple sclerosis have also been shown to experience clinical improvement through NK cell expansions (*Segal, 2007*). NK cells have also been shown to exert a protective role in brain by removing viral infection and activated microglia (*Shi et al., 2011*).

We also found that higher Th2 differentiation of CD4[+] T cells was associated with a lower risk of death after ALS diagnosis, corroborating previous findings of the neuroprotective role of CD4[+] T cells in ALS (*Beers et al., 2008*; *Jones et al., 2015*). Animal studies have suggested Th1 and Th17 cells to promote neuroinflammation by producing proinflammatory cytokines and enhancing microglia-mediated neurotoxic effects (*Murphy et al., 2010*) whereas Th2 cells and regulatory T cells (Tregs) to suppress neuroinflammation by producing anti-inflammatory cytokines and enhancing microglia-mediated neuroprotective effects (*Gendelman and Appel, 2011*; *Beers et al., 2011*). Although not statistically significant, our study indeed found a trend for Th1-differentiatated CD4[+] CM cells and Th17-differentiated CD4[+] CM cells to be positively associated with the risk of death after ALS diagnosis.

A novel finding of the present study is that higher proportions of CD8[+] T cells and CD4[+] EMRA T cells were associated with a higher risk of death after ALS diagnosis. The precise underlying mechanisms linking together these cell types with ALS survival are unclear. However, previous animal study showed that activated CD8[+] T cells were present in CNS of *SOD1^{G93A}* ALS model at the symptomatic stage, and that selective deletion of CD8[+] T cells could increase the survival of motoneurons whereas coculture motoneurons with mutant SOD1-expressing CD8[+] T lymphocytes could selectively kill the motoneurons via Fas and granzyme pathways (*Coque et al., 2019*). Previous studies also indicated that CD4[+] EMRA cells could demonstrate cytotoxic features and express the chemokine receptor CX3CR1 in the setting of Dengue virus infection (*Weiskopf et al., 2015*), which are associated with cytotoxic lymphocytes with cytoplasmic granules containing perforin and granzymes (*Nishimura et al., 2002*). Further studies are clearly needed to study more in detail the functional relevance of CD8[+] and CD4[+] EMRA T cells in ALS.

Expansions in *C9orf72*, the most common genetic cause of ALS, have been shown to be associated with immune features (*Lai and Ichida, 2019*), including activation of microglia and elevated levels of peripheral inflammatory cytokines (*Trageser et al., 2019*). In our study, although patients with or without *C9orf72* expansions did not differ greatly in terms of major leukocyte populations, patients with *C9orf72* expansions appeared to display changes in certain lymphocyte subpopulations including NK, T, naive CD4[+] T, and CD8[+] EM cells compared with patients without such expansions. Because of the relatively small number of patients with *C9orf72* expansions in the study, these results should however be interpreted with caution until validated further.

Sex-based immunological difference might exist in response to external and internal antigens, contributing potentially to the variations in the incidence of autoimmune diseases and malignancies as well as the difference in response to vaccines between men and women (*Klein and Flanagan, 2016*). Murdock et al. also demonstrated sex-specific effect of NK cells and neutrophils in ALS (*Murdock et al., 2021a*; *Murdock et al., 2021b*). In the present study, we found that men and women showed different longitudinal trajectories of some lymphocyte subpopulations but not leukocyte populations. Although these findings need to be validated, they add on the evidence base to support a potential sex-specific immune response in ALS and ALS as a heterogenous disease.

Our study is the first, to our knowledge, to report the role of a comprehensive list of immune cells, including neutrophils, lymphocytes, monocytes, and detailed T-cell phenotypes, on the prognosis of ALS. The strengths of the present study are the large number of ALS patients which were representative of all ALS patients in the source population, the rich information on disease characteristics including genetic causes, the long (up to 5 years after diagnosis) and complete follow-up (through the MND Quality Registry), the availability of both routine cell counts and detailed lymphocyte phenotyping, as well as the access to repeated measures over time. The study also has limitations. First, the main cohort was heterogeneous in terms of the numbers of cell measurements and the time intervals between measurements, as the timing of blood sampling was not predefined. Indication

bias due to, for example, ongoing infections might therefore be a concern. The sensitivity analysis excluding all samples taken at the time of infections provided however rather similar results. Except for infection, other conditions such as immune diseases and allergies could also affect immune cell populations and should ideally also have been taken into account in the analysis. Further, the longitudinal analysis of cell counts should be interpreted with caution because not all patients contributed repeated cell measurements. This is however an unavoidable problem for any longitudinal study of ALS patients, given the high mortality rate of this patient group. Regardless, when focusing on the first cell measures, we obtained similar results as in the main analysis. It would therefore be interesting to compare ALS with other diseases, especially other neurodegenerative diseases, regarding the studied cell counts, in terms of both their longitudinal trajectories during disease course and their prognostic values in predicting patient outcome. Further, the FITMaN panel does not include B-cell subtypes or Treg cells, which await to be studied further. Finally, causal inferences on the functional implications of the reported associations are not possible due to the observational study design.

In conclusion, our findings suggest a dual role of immune responses in ALS prognosis, where neutrophils and monocytes primarily reflect functional status whereas different T lymphocyte populations act as prognostic markers for survival. These findings provide additional insights for cell-based therapy in prolonging survival in ALS.

## Materials and methods
### Study cohort
The Swedish Motor Neuron Disease (MND) Quality Registry was established in 2015, collecting information on clinical characteristics, biological test results, and quality of life outcomes from >80% of MND patients in Sweden (*Longinetti et al., 2018*). Since 2017, the MND Quality Registry has included 99% of MND patients in the Stockholm area among whom 97.1% are diagnosed with ALS. All ALS diagnoses were made by a specialist in neurology and followed up by a neuromuscular specialist, and met the diagnostic requirement of definite, probable, probable laboratory supported, or possible ALS according to the revised El Escorial criteria (*Ludolph et al., 2015*; *Brooks et al., 2000*). To ensure the accuracy of diagnosis, all patients in the registry are re-evaluated annually to update diagnosis, whenever needed.

Through the MND Quality Registry, we first identified 420 patients with ALS diagnosed from the start of the registry until October 7, 2020, in Stockholm. We reviewed the medical records of these patients to identify information on peripheral leukocyte populations (i.e., differential leukocyte counts). During this process, we excluded 12 patients who were not diagnosed at the ALS Research and Care Center, Karolinska University Hospital – the only tertiary referral center for ALS in Stockholm, three patients with unknown time of symptom onset, 82 patients lacking leukocyte counts, and 35 patients with counts outside of the stipulated observation period (i.e., from 3 months before date of diagnosis until October 7, 2020). The final analysis cohort included 288 patients (68.6%), with at least one recorded differential leukocyte count during the observation period. The included patients did not differ significantly in terms of demographic and clinical characteristics from the excluded patients (*Table 4*). About half of the patients had a single measurement of leukocytes whereas the other half had been sampled two or more times.

Among the ALS patients diagnosed at the Karolinska University Hospital and with a date of symptom onset ($N$ = 405), we further performed flow cytometry in 92 patients ('FlowC cohort') to determine lymphocyte subpopulations (i.e., T, B, and NK cells) as well as an extended T lymphocyte panel 'FITMaN' – an internationally standardized panel reported by the Flow Immunophenotyping Technical Meeting at NIH (*Maecker et al., 2012*). Compared to the main study cohort, patients of the FlowC cohort were slightly younger and more likely to have a limb onset (*Supplementary file 1*). In both the main and FlowC cohorts, we followed the ALS patients from date of diagnosis or first cell measurement (differential leukocyte counts or FlowC), whichever came later, until occurrence of the outcome of interest (i.e., death or use of invasive ventilation) or October 7, 2020, whichever came first.

### Blood samples and flow cytometric analysis
All samples were freshly collected. The sample processing and analyzing procedures were according to the validated protocol at the Departments of Clinical Chemistry (differential leukocyte counts) and

**Table 4.** Characteristics of the 288 patients with amyotrophic lateral sclerosis (ALS) included in the study, compared with the entire population of ALS patients during the study period in Stockholm, Sweden.

| Characteristics | Patients included in the study (*N* = 288) | All patients in Stockholm (*N* = 420) | p value for difference[*] |
|---|---|---|---|
| Sex, *N* (%) | | | 0.48 |
| Female | 134 (47%) | 201 (48%) | |
| Male | 154 (53%) | 219 (52%) | |
| Age at diagnosis, years | | | 0.02 |
| Median (Q1, Q3) | 65 (56, 71) | 66 (57, 72) | |
| Diagnostic delay, months | | | 0.94 |
| Median (Q1, Q3) | 12.30 (7.88, 19.93) | 12.35 (7.59, 20.54) | |
| Gene mutation, *N* (%)[†] | | | 1.00 |
| *SOD1* | 7 (2.88%) | 9 (2.56%) | |
| *C9orf72* | 22 (9.05%) | 30 (8.55%) | |
| Other | 4 (1.65%) | 5 (1.42%) | |
| Site of onset, *N* (%) | | | 0.26 |
| Limb | 182 (63%) | 250 (60%) | |
| Bulbar | 78 (27%) | 118 (28%) | |
| Other | 20 (7%) | 32 (8%) | |
| Missing | 8 (3%) | 20 (5%) | |
| Family history, *N* (%) | | | 0.19 |
| Yes | 19 (7%) | 30 (7%) | |
| No | 144 (50%) | 201 (48%) | |
| Not clear | 3 (1%) | 7 (2%) | |
| Missing | 122 (42%) | 182 (43%) | |
| No. of measurements for cell count (%) | | | – |
| One | 146 (51%) | – | |
| Two | 75 (26%) | – | |
| Three | 35 (12%) | – | |
| Four or more | 32 (11%) | – | |

[*]p value for the differences between patients included in the study and patients not included in the study; Wilcoxon rank sum test was used for the comparison of continuous variables whereas chi-square test was used for the comparison of categorical variables.

[†]Results available for 243 of the 288 patients included in the study, and 351 of the entire 420 patients in Stockholm.

Clinical Immunology and Transfusion Medicine (FlowC), Karolinska University Hospital. All analyses were performed during daytime, within 24 hr of sampling. Differential leukocyte counts were done on a Sysmex XN-9000 (Sysmex, Kobe, Japan). FlowC was implemented in clinical routine based on the standardized phenotyping panel by the Human Immunophenotyping Consortium with a set of defined 8-color antibody cocktails (*Maecker et al., 2012*). The experiments were performed on a triple-laser Beckman Coulter Gallios and analyzed by Kaluza Software (Beckman Coulter, Brea, CA).

Differential leukocyte counts included neutrophils, lymphocytes, monocytes, eosinophils, and basophils. We did not include eosinophils and basophils in the analysis as they in most cases were low to undetectable. In addition to studying leukocytes individually, we also analyzed NLR as suggested

by previous studies (*Choi et al., 2020*; *Wei et al., 2022*). The FITMaN panel included measures of 23 lymphocyte subpopulations, including (1) counts of B cells, NK cells, and T cells; (2) %s of CD4$^+$ and CD8$^+$ T cell subtypes (i.e., naive, CM, EM, and effector memory cells re-expressing CD45RA [EMRA] T cells based on CCR7 and CD45RA expression, as well as Th1, Th2, and Th17 of CM and EM CD4$^+$ T cells based on CXCR3 and CCR6 expression); and (3) %s of subtypes of activated CD4$^+$ and CD8$^+$ T cells based on the expression of HLA-DR and CD38 (i.e., CD4$^+$HLA-DR$^+$CD38$^-$ cells, CD4$^+$HLA-DR$^+$CD38$^+$ cells, CD8$^+$HLA-DR$^+$CD38$^-$ cells, and CD8$^+$HLA-DR$^+$CD38$^+$ cells). T cells were gated from a lymphocyte (FCS/SSC) gate as cells expressing CD45 and CD3. The unit of cell count was 10$^9$/l whereas the %s were expressed as the proportions of the immune cell populations out of their parent populations. The normal references for leukocytes (main cohort) were reported by the Department of Clinical Chemistry, Karolinska University Hospital, based on the Nordic Reference Interval Project (NORIP) (*Nordin et al., 2004*) (leukocyte: 3.5–8.8; neutrophil: 1.6–5.9; lymphocyte: 1.1–3.5; monocyte: 0.2–0.8; 10$^9$/l). The reference values for FlowC were reported in the form of 5th to 95th percentiles using reference normal ranges obtained from 50 healthy adults. All cell counts and %s were retrieved from patient medical records.

## Outcomes of interest

The primary study outcome was risk of death or use of invasive ventilation after ALS diagnosis, identified from the MND Quality Registry. The secondary study outcomes included functional status measured through the ALSFRS-R and disease progression rate. ALSFRS-R is a questionnaire-based scale that measures the motor function and disease severity of ALS patients and is considered the gold standard measure of disability progression (*Makary et al., 2021*). Higher ALSFRS-R score indicates better functional status. We acquired information on all available ALSFRS-R scores for the ALS patients from the MND Quality Registry. Progression rate measures the rate of ALSFRS-R decline and were calculated by dividing the difference between 48 (the full score) and measured ALSFRS-R score at a specific time point by the time difference between time of symptom onset to the measurement time of ALSFRS-R (in months). Progression rate is an independent prognostic predictor for ALS (*Labra et al., 2016*).

## Other clinical characteristics

Data on sex, age at diagnosis, diagnostic delay, gene mutation, site of onset, family history, and body mass index (BMI) were collected from the MND Quality Registry or medical records. Diagnostic delay was calculated as time difference between date of onset and date of diagnosis. Site of onset was categorized as 'limb', 'bulbar', 'other', and 'missing'. The definition of family history was based on whether a clear history of ALS existed among the relatives and categorized as 'yes', 'no', 'not clear', and 'missing'. Genetic testing was offered to all patients with ALS around the time of diagnosis at the ALS Research and Care Center at Karolinska University Hospital. A total of 88 of the most common ALS-contributing genes, including *SOD1*, *C9orf72*, *FUS*, *TARDBP*, *TBK1*, *OPTN*, *VCP*, etc., were screened.

## Statistical analysis

To better understand the studied cell populations in ALS, we first performed a few analyses focusing on the different cell populations alone. We first calculated the mean levels of measured cell populations among ALS patients, by summarizing all measurements from 3 months before diagnosis until end of follow-up. To visualize the temporal patterns of the cell populations, we drew a trajectory line of all measurements for each cell type and each patient. We then used the locally estimated scatterplot smoothing curves with 95% CIs to show the temporal pattern of the predicted median level of each cell type after ALS diagnosis. If the CIs did not overlap with the normal ranges of the cell populations, we considered the observed levels among ALS patients to be statistically deviant from normal ranges. We also used linear mixed model to assess the within-individual temporal changes of cell populations after ALS diagnosis. In this analysis, we included a random intercept to account for the initial differences between individuals and adjusted for age at diagnosis and sex. We analyzed all ALS patients together first and then separately by sex, site of onset and presence of *C9orf72* expansions. Patients with *C9orf72* expansions have been suggested to demonstrate a different immune phenotype compared with ALS patients without such expansions (*Pinilla et al.,*

*2021*; *McCauley et al., 2020*). To evaluate whether ALS treatment would influence the cell counts, we further visualized the temporal patterns of differential leukocyte counts before and after Riluzole treatment.

We next used Cox model to derive HR and 95% CI to assess the association of different cell populations with the risk of death, after adjustment for other prognostic indicators of ALS including age at diagnosis, sex, site of onset, diagnostic delay, ALSFRS-R score, time difference between the measure of ALSFRS-R score and diagnosis, BMI, and time difference between the measure of BMI and diagnosis. We used time since diagnosis as the underlying time scale and cluster robust variance estimation to account for dependence of repeated measurements. In this analysis, we log-transformed the counts of leukocytes, neutrophils, lymphocytes, and monocytes, as well as %s of CD4$^+$ EM cells, CD4$^+$ EMRA cells, Th2 of CD4$^+$ EM cells, Th17 of CD4$^+$ EM cells, Th2 of CD4$^+$ CM cells, CD8$^+$ T cells, CD8$^+$ CM cells, CD4$^+$HLA-DR$^+$CD38$^-$ cells, CD4$^+$HLA-DR$^+$CD38$^+$ cells, CD8$^+$HLA-DR$^+$CD38$^-$ cells, and CD8$^+$HLA-DR$^+$CD38$^+$ cells, to achieve a better normal distribution. Values of other cell populations were used as is (i.e., without transformation). For all markers, we estimated the effect size per SD increase. Because some patients had their first cell measurements after diagnosis, we also took into account left truncation in all analyses. Eight patients (3%) had missing values on site of onset and were excluded in the multivariable analysis where site of onset was adjusted for.

The proportional hazards assumption of the Cox model was assessed using the Schoenfeld residual test. After stratifying the analysis by site of onset which deviated from the assumption, the assumption became satisfied for all other variables. As HRs obtained in the stratified analysis were nearly the same as those obtained without stratification, for simplicity and consistency, we reported all findings from the original models (without stratification by site of onset).

Among the 288 patients, 57 (19.8%) were diagnosed before the start of the MND Quality Registry. In a sensitivity analysis, we excluded these patients with the aim to see if the results obtained in the main analysis would pertain to a cohort of incident patients. This analysis was not performed for the FlowC cohort as vast majority (*N* = 85) of the 92 patients were incident patients. We also conducted a sensitivity analysis by only focusing on the first cell measure of each patient to examine whether the main results would differ after removal of repeated measurements. We then performed another sensitivity analysis by excluding patients with *C9orf72* expansions, to understand the influence of *C9orf72* expansions on the results.

We further used a generalized estimating equation (GEE) model to assess the correlations of cell populations with functional status (i.e., ALSFRS-R score) and disease progression rate measured at the same time as the cell markers. The GEE model considers the correlations between repeated measurements within the same individual. Like the survival analysis above, we used log-transformed values for some cell measures whereas original values for others and estimated the effect size per SD increase of the cell measures, after adjustment for age at diagnosis and sex. Additionally, we also used the GEE model to assess the correlations between longitudinal changes in differential leukocyte counts and the longitudinal changes in ALSFRS-R score.

Finally, to assess whether the results on cell measures and the study outcomes would be influenced by ongoing infections, we performed another sensitivity analysis after excluding all measurements taken during an ongoing infection. This was done by reviewing medical records of all patients to see if at the time of blood sampling the patient had a high count of leukocyte (>8.8 × 10$^9$/l), a diagnosis of any infectious disease, specific tests for infection, or self-reported infectious symptoms.

All analyses were performed using R 3.6.2. A two-sided p value of <0.05 was considered statistically significant. To correct for multiple testing, we also computed the Benjamini–Hochberg false discovery rate (*Pawitan et al., 2005*).

## Acknowledgements

This work was supported by the European Research Council (ERC) Starting Grant (MegaALS, No. 802091), the Swedish Research Council (No. 2019-01088), Karolinska Institutet (Strategic Research Area in Epidemiology and Senior Researcher Award), and China Scholarship Council. The funders had no role in the design of the study and collection, analysis, and interpretation of data and in writing the manuscript.

## Additional information

### Funding

| Funder | Grant reference number | Author |
|---|---|---|
| European Research Council | Starting Grant | Fang Fang |
| Swedish Research Council | No. 2019-01088 | Fang Fang |
| Karolinska Institutet | Strategic Research Area in Epidemiology and Senior Researcher Award | Fang Fang |
| China Scholarship Council | 201700260293 | Can Cui |
| European Research Council | MegaALS, No. 802091 | Fang Fang |

The funders had no role in study design, data collection, and interpretation, or the decision to submit the work for publication.

### Author contributions

Can Cui, Conceptualization, Data curation, Formal analysis, Funding acquisition, Methodology, Visualization, Writing - original draft, Writing - review and editing; Caroline Ingre, Conceptualization, Resources, Writing - review and editing; Li Yin, Methodology, Supervision, Validation, Writing - review and editing; Xia Li, Methodology, Writing - review and editing; John Andersson, Supervision, Writing - review and editing; Christina Seitz, Nicolas Ruffin, Writing - review and editing; Yudi Pawitan, Conceptualization, Methodology, Supervision, Writing - review and editing; Fredrik Piehl, Conceptualization, Supervision, Writing - review and editing; Fang Fang, Conceptualization, Funding acquisition, Methodology, Supervision, Writing - review and editing

### Author ORCIDs

Can Cui http://orcid.org/0000-0001-8138-0495
Xia Li http://orcid.org/0000-0003-1922-7152
John Andersson http://orcid.org/0000-0003-2799-6349
Nicolas Ruffin http://orcid.org/0000-0002-3698-5505
Fang Fang http://orcid.org/0000-0002-3310-6456

### Ethics

EthicsThe study was approved by the Regional Ethical Review Board in Stockholm (reference number: 2017/1895-31/1). Because of the register-based nature of the study, the requirement of informed consent was waived.

### Decision letter and Author response

Decision letter https://doi.org/10.7554/eLife.74065.sa1
Author response https://doi.org/10.7554/eLife.74065.sa2

## Additional files

### Supplementary files

• Supplementary file 1. Characteristics of the 92 patients with amyotrophic lateral sclerosis (ALS) included in the analysis of FlowC test, compared with the entire population of ALS patients during the study period in Stockholm, Sweden.

• Supplementary file 2. Mean levels of leukocyte subpopulations ($N$ = 288 patients) and lymphocyte subpopulations ($N$ = 92 patients) across all measures.

• Supplementary file 3. Temporal changes of lymphocyte populations after diagnosis of amyotrophic lateral sclerosis (ALS), analysis of 92 patients with FlowC test.

• Supplementary file 4. Associations of leukocyte populations with the risk of death after a diagnosis of amyotrophic lateral sclerosis (ALS), a cohort study of 288 patients with ALS in Stockholm, Sweden.

• Supplementary file 5. Sensitivity analyses of the associations of leukocyte populations with risk

of death after a diagnosis of amyotrophic lateral sclerosis (ALS), focusing on newly diagnosed ALS patients, first cell measure only, or excluding patients with C9orf72 expansions*.

• Supplementary file 6. Cross-sectional correlations between lymphocyte populations and Amyotrophic Lateral Sclerosis Functional Rating Scale – revised (ALSFRS-R) score and disease progression rate, a cohort study of 92 ALS patients in Stockholm, Sweden*.

• Supplementary file 7. Sensitivity analyses of the associations of leukocyte populations with Amyotrophic Lateral Sclerosis Functional Rating Scale – revised (ALSFRS-R) score and disease progression rate, after removing the the blood samples with potential ongoing infection*.

• Supplementary file 8. Sensitivity analyses of associations of leukocyte populations with the risk of death after a diagnosis of amyotrophic lateral sclerosis (ALS), after removing the blood samples with potential ongoing infection*.

• Transparent reporting form

• Source code 1. Source code for Table 1 and 2.

### Data availability

The original data are held by the Swedish Motor Neuron Disease Quality Registry and are not publicly available due to Swedish laws. However, any researcher who is interested can access the data by obtaining an ethical approval from regional ethical review board. Detailed information of the Swedish Ethical Review Authority can be found at https://etikprovningsmyndigheten.se. Anonymized data that support the findings of this study are available on reasonable request from the corresponding authors, in agreement with European regulations and Regional Ethical Review Board in Sweden. One source file containing R code (Source code 1) for table 1 and 2 has been provided.

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
