## [Editor Report]

Cui et al. colleagues carried out a longitudinal analysis of blood cell counts in a cohort of ALS patients and found increased numbers of neutrophils and monocytes, which negatively correlated with ALSFRS-R score, but not with rate of disease progression. They also found increased levels in NK and central memory TH2 T cells, which correlated with a lower risk of death. In contrast, increased levels of CD4 CD45RA effector memory and CD8 T cells were correlated with a higher risk of death. These findings have important implications for the pathogenesis of ALS as well as the development of immune-based ALS therapies targeting specific populations immune cells.

---

## [Decision Letter]

**Decision letter after peer review:**

Thank you for submitting your article "Correlation between leukocyte phenotypes and prognosis of amyotrophic lateral sclerosis: a longitudinal cohort study" for consideration by *eLife*. Your article has been reviewed by 2 peer reviewers, and the evaluation has been overseen by a Reviewing Editor and Catherine Dulac as the Senior Editor. The reviewers have opted to remain anonymous.

Essential revisions:

1) Please address:

"Given these issues, one really would want to see disease controls, and how the different cell counts change in another disease. Finally, there is not discussion about how or whether treatments, or changes in treatment, could influence observed counts. "

2. "Levels of cell counts correlate with ALSFRSR but not disease progression. It would be interesting to know whether the change in cell counts correlate with changes in ALSFRSR."

3) "The description of primary outcome is confusing, as this is typically referring to a single prespecified outcome and associated statistical analysis, whereas the paper initially focuses on longitudinal changes in cell counts before asking how difference in a large number of cell sets were or were not predictive of survival."

4) "Sex has been identified as a determinant of immune system profiles in ALS (see PMIDs 33974561, 33531377, 32388062, 33715827, for example). Although the authors investigated the impact of C9orf72 and segment onset, it would be interesting to determine whether sex impacted immune profiles in this cohort."

5) "A couple of studies have recently reported on the prognostic role of neutrophil-to-lymphocyte ratio in cross-sectional studies and in Asian ALS populations (PMIDs 31949271, 34472488). It would be insightful to determine whether these findings are replicated here."

*Reviewer #1 (Recommendations for the authors):*

Levels of cell counts correlate with ALSFRSR but not disease progression. It would be interesting to know whether the change in cell counts correlate with changes in ALSFRSR.

The description of primary outcome is confusing, as this is typically referring to a single prespecified outcome and associated statistical analysis, whereas the paper initially focuses on longitudinal changes in cell counts before asking how difference in a large number of cell sets were or were not predictive of survival.

*Reviewer #2 (Recommendations for the authors):*

(1) Sex has been identified as a determinant of immune system profiles in ALS (see PMIDs 33974561, 33531377, 32388062, 33715827, for example). Although the authors investigated the impact of C9orf72 and segment onset, it would be interesting to determine whether sex impacted immune profiles in this cohort.

(2) A couple of studies have recently reported on the prognostic role of neutrophil-to-lymphocyte ratio in cross-sectional studies and in Asian ALS populations (PMIDs 31949271, 34472488). It would be insightful to determine whether these findings are replicated here.

(3) The PMIDs 31949271, 34472488, 33974561, 33531377 are missing from the discussion.

(4) Additionally, T cells have been investigated in ALS: although a genetic model, the study by Coque et al. is very salient (PMID 30674678), in sporadic ALS patients (PMID 34405141).

---

## [Author Response]

Essential revisions:1) Please address:"Given these issues, one really would want to see disease controls, and how the different cell counts change in another disease. Finally, there is not discussion about how or whether treatments, or changes in treatment, could influence observed counts. "

Regarding disease control, we have now shown a figure about longitudinal changes of the studied cell populations in a cohort of 34 patients with multiple sclerosis measured during the same time period using the identical methodology at the Karolinska University Hospital.

Regarding treatment, as a vast majority (89.6%) of the ALS patients included in the study were treated with Riluzole, we have now added an analysis to show the cell counts before and after Riluzole treatment in the manuscript. Please see response to Reviewer 1 below.

2. "Levels of cell counts correlate with ALSFRSR but not disease progression. It would be interesting to know whether the change in cell counts correlate with changes in ALSFRSR."

We have now added the suggested analysis in the manuscript. In brief, we found that a greater increase of leukocytes, neutrophils and monocytes was associated with a greater decline in ALSFRS-R score.

3) "The description of primary outcome is confusing, as this is typically referring to a single prespecified outcome and associated statistical analysis, whereas the paper initially focuses on longitudinal changes in cell counts before asking how difference in a large number of cell sets were or were not predictive of survival."

The primary outcome of the study is risk of death (alternatively use of invasive ventilation). The secondary outcomes are ALSFRS-R score and disease progression rate. The longitudinal changes of cell counts were described to understand the temporal changes of the exposure (i.e., cell counts). We have now clarified these in the manuscript.

4) "Sex has been identified as a determinant of immune system profiles in ALS (see PMIDs 33974561, 33531377, 32388062, 33715827, for example). Although the authors investigated the impact of C9orf72 and segment onset, it would be interesting to determine whether sex impacted immune profiles in this cohort."

We have now added a figure to show the sex-specific longitudinal changes of cell counts.

5) "A couple of studies have recently reported on the prognostic role of neutrophil-to-lymphocyte ratio in cross-sectional studies and in Asian ALS populations (PMIDs 31949271, 34472488). It would be insightful to determine whether these findings are replicated here."

We have now added analysis for neutrophil-to-lymphocyte ratio in the manuscript. Please see response to Reviewer 2 below.

Reviewer #1 (Recommendations for the authors):Levels of cell counts correlate with ALSFRSR but not disease progression. It would be interesting to know whether the change in cell counts correlate with changes in ALSFRSR.

Thank you for the good suggestion. We have now used generalized estimating equation model to evaluate the associations between changes in cell measures and changes in ALSFRS-R score (see Table 4). In brief, the results showed that a greater longitudinal increase of leukocyte, neutrophils and monocyte was associated with a greater longitudinal decline in ALSFRS-R score.

We have now added this new analysis in Methods and Results.

Page 17, paragraph 2:

“Additionally, we also used the GEE model to assess the correlations between longitudinal changes in differential leukocyte counts and the longitudinal changes in ALSFRS-R score.”

Page 6, paragraph 4:

“A greater longitudinal increase of leukocytes, neutrophils and monocytes was associated with a greater longitudinal decline in ALSFRS-R score (Table 4).”

The description of primary outcome is confusing, as this is typically referring to a single prespecified outcome and associated statistical analysis, whereas the paper initially focuses on longitudinal changes in cell counts before asking how difference in a large number of cell sets were or were not predictive of survival.

The primary outcome of the study is risk of death (alternatively use of invasive ventilation). The secondary outcomes are ALSFRS score and disease progression rate. The longitudinal changes of cell counts were described to show the temporal changes of the exposure (i.e., cell measurements). We have now clarified these in the manuscript.

Page 14, paragraph 2:

“The primary study outcome was risk of death or use of invasive ventilation after ALS diagnosis, identified from the MND Quality Registry. The secondary study outcomes included functional status measured through the Amyotrophic Lateral Sclerosis Functional Rating Scale – revised (ALSFRS-R) and disease progression rate.”

Page 15, paragraph 3:

“To better understand the studied cell populations in ALS, we first performed a few analyses focusing on the different cell populations alone. […] We then used the locally estimated scatterplot smoothing (LOESS) curves with 95% confidence intervals (CIs) to show the temporal pattern of the predicted median level of each cell type after ALS diagnosis…”

Reviewer #2 (Recommendations for the authors):(1) Sex has been identified as a determinant of immune system profiles in ALS (see PMIDs 33974561, 33531377, 32388062, 33715827, for example). Although the authors investigated the impact of C9orf72 and segment onset, it would be interesting to determine whether sex impacted immune profiles in this cohort.

Thank you for the suggestion. We have now performed analysis to show the sex-specific longitudinal changes of the cell populations and added this new analysis to Methods and Results, including a new panel to Figure 1—figure supplement 1 and a new Figure 1—figure supplement 4.

Page 15, paragraph 3:

“We analyzed all ALS patients together first and then separately by sex, site of onset and presence of C9orf72 expansions.”

Page 5, paragraph 2:

“These results remained largely similar when stratifying the patients by sex, site of onset or presence of C9orf72 expansions (Figure 1—figure supplement 1).”

Page 5, paragraph 3:

“Male patients showed lower levels of CD4^+^, naïve CD4^+^ and Th2 of CD4^+^ CM cells, but higher levels of CD4^+^ EM, CD4^+^ EMRA, CD8^+^, CD4^+^HLA-DR^+^CD38^-^, and CD4^+^HLA-DR^+^CD38^+^ cells, compared with female patients, especially early stage after diagnosis (Figure 1—figure supplement 4).”

(2) A couple of studies have recently reported on the prognostic role of neutrophil-to-lymphocyte ratio in cross-sectional studies and in Asian ALS populations (PMIDs 31949271, 34472488). It would be insightful to determine whether these findings are replicated here.

Thank you for the suggestion. In the study by Choi et al. (PMID 31949271), ALS patients with the highest (above 3rd tertile) baseline neutrophil-to-lymphocyte ratio (NLR) were found to have a higher risk of death compared with patients with the lowest (below 1st tertile) baseline NLR (HR=1.60, 95%CI: 1.01–2.51, P = 0.041). However, no difference in disease progression rate was found between patients with different levels of NLR. In the study by Wei et al. (PMID 34472488), 1030 ALS patients were divided into three groups according to NLR: Group 1 (NLR <2), Group 2 (NLR 2–3), and Group 3 (NLR >3). The survival time differed between these groups with Group 3 showing the shortest survival. A 1-unit increase in NLR was associated with a higher mortality (HR=1.079, 95% CI: 1.016–1.146, P=0.014) whereas disease progression rate was also found to differ between the three groups.

We have now analyzed the association between NLR and risk of death in our study, and found that an increase in NLR was indeed associated with a higher risk of death (HR=1.31, 95% CI: 1.13-1.52 per SD increase in NLR). We also found that disease progression rate differed between patients with different NLRs (e.g., p=0.04, comparing group with above the 3^rd^ tertile NLR to the group with below the 1^st^ tertile NLR). We have now added this new analysis to Methods, Results, and Discussion (including citing the two references).

Page 13, paragraph 4:

“In addition to studying leukocytes individually, we also analyzed neutrophil-to-lymphocyte ratio (NLR) as suggested by previous studies^23,24^.”

Page 6, paragraph 2:

“Per SD increase of NLR, there was a higher risk of death (HR=1.31, 95% CI: 1.13-1.52).”

Page 6, paragraph 4:

“Disease progression rate differed however between patients with different NLRs (e.g., p=0.04, comparing group with above 3^rd^ tertile NLR to the group with below 1^st^ tertile NLR).”

Page 8, paragraph 2:

“NLR has been considered a prognostic marker in many diseases, such as cancer^20,21^ and cardiovascular disease^22^, which may indicate the involvement of complex coordination between different immune response pathways in the disease progression. […] However, the relation between NLR and disease progression rate of ALS is less consistently reported between studies^23,24^, although we did find a higher NLR to be associated with a greater disease progression rate.”

(3) The PMIDs 31949271, 34472488, 33974561, 33531377 are missing from the discussion.

Thank you. We have now added these references in the Discussion.

Page 8, paragraph 2:

“NLR has been considered a prognostic marker in many diseases, such as cancer^20,21^ and cardiovascular disease^22^, which may indicate the involvement of complex coordination between different immune response pathways in the disease progression. […] However, the relation between NLR and disease progression rate of ALS is less consistently reported between studies^23,24^, although we did find a higher NLR to be associated with a greater disease progression rate.”

Page 10, paragraph 3:

“Sex-based immunological difference might exist in response to external and internal antigens, contributing potentially to the variations in the incidence of autoimmune diseases and malignancies as well as the difference in response to vaccines between men and women^44^. […] Although these findings need to be validated, they add on the evidence base to support a potential sex-specific immune response in ALS and ALS as a heterogenous disease.”

(4) Additionally, T cells have been investigated in ALS: although a genetic model, the study by Coque et al. is very salient (PMID 30674678), in sporadic ALS patients (PMID 34405141).

We have now cited the two studies in the Discussion.

Page 9, paragraph 1:

“A previous study showed that lower levels of CD56^bright^ NK cells in the CSF were associated with faster progression in ALS patients^29^. Additionally, findings from experimental autoimmune encephalomyelitis…”

In the original manuscript (page 9, paragraph 3), we stated “Previously, CD8^+^ T cells have been found to be present in the spinal cord only at the end stage of ALS^5^. However, such cells have been suggested to have a pathogenic potential in multiple sclerosis and to communicate with mononuclear phagocytes^47,48^.”

We have now revised these sentences as “However, previous animal study showed that activated CD8^+^ T cells were present in the CNS of SOD1^G93A^ ALS model at the symptomatic stage, and that selective deletion of CD8^+^ T cells could increase the survival of motoneurons whereas coculture motoneurons with mutant SOD1-expressing CD8^+^ T lymphocytes could selectively kill the motoneurons via Fas and granzyme pathways^37^.”